# Temporal Trends in Treatment and Outcomes of Endometrial Carcinoma in the United States, 2005–2020

**DOI:** 10.3390/cancers16071282

**Published:** 2024-03-26

**Authors:** Victor Adekanmbi, Fangjian Guo, Christine D. Hsu, Daoqi Gao, Efstathia Polychronopoulou, Itunu Sokale, Yong-Fang Kuo, Abbey B. Berenson

**Affiliations:** 1Center for Interdisciplinary Research in Women’s Health, School of Medicine, The University of Texas Medical Branch, Galveston, TX 77555-0587, USA; faguo@utmb.edu (F.G.); cdhsu@utmb.edu (C.D.H.); abberens@utmb.edu (A.B.B.); 2Department of Obstetrics and Gynecology, The University of Texas Medical Branch, Galveston, TX 77555-0587, USA; 3Department of Biostatistics and Data Science, The University of Texas Medical Branch, Galveston, TX 77555-0587, USA; dagao@utmb.edu (D.G.); efpolych@utmb.edu (E.P.); yokuo@utmb.edu (Y.-F.K.); 4Department of Medicine, Section of Epidemiology and Population Sciences, Baylor College of Medicine, Houston, TX 77030, USA; itunu.sokale@bcm.edu; 5Dan L. Duncan Comprehensive Cancer Center, Baylor College of Medicine, Houston, TX 77054, USA

**Keywords:** trends, characteristics, treatment, outcomes, endometrial carcinoma

## Abstract

**Simple Summary:**

Endometrial carcinoma is the most frequently diagnosed gynecologic cancer in the United States. It used to be known as a disease of only postmenopausal women. Currently, pre-menopausal women are being diagnosed with endometrial cancer at a higher rate; however, this condition is primarily a disease of post-menopausal women. Hysterectomy, bilateral salpingo-oophorectomy, and pelvic lymph node dissection followed by adjuvant therapy have been the standard treatment of endometrial carcinoma. Management of advanced endometrial carcinoma has become more advanced and individualized. The introduction of minimally invasive surgery, immunotherapy, and neoadjuvant chemotherapy as treatment approaches has resulted in improved short- to long-term outcomes. The results of this study indicate that, despite an increase in the number of individuals with advanced endometrial carcinoma, the overall survival rate has improved significantly, probably due to advancements in treatment options and patient care.

**Abstract:**

Endometrial cancer has continued to see a rising incidence in the US over the years. The main aim of this study was to assess current trends in patients’ characteristics and outcomes of treatment for endometrial carcinoma over 16 years. A dataset from the National Cancer Database (NCDB) for patients diagnosed with endometrial carcinoma from 2005 to 2020 was used in this retrospective, case series study. The main outcomes and measures of interest included tumor characteristics, hospitalization, treatments, mortality, and overall survival. Then, 569,817 patients who were diagnosed with endometrial carcinoma were included in this study. The mean (SD) age at diagnosis was 62.7 (11.6) years, but 66,184 patients (11.6%) were younger than 50 years, indicating that more patients are getting diagnosed at younger ages. Of the patients studied, 37,079 (6.3%) were Hispanic, 52,801 (9.3%) were non-Hispanic Black, 432,058 (75.8%) were non-Hispanic White, and 48,879 (8.6%) were other non-Hispanic. Patients in the 4th period from 2017 to 2020 were diagnosed more with stage IV (7.1% vs. 5.2% vs. 5.4% vs. 5.9%; *p* < 0.001) disease compared with those in the other three periods. More patients with severe comorbidities (Charlson Comorbidity Index score of three) were seen in period 4 compared to the first three periods (3.9% vs. ≤1.9%). Systemic chemotherapy use (14.1% vs. 17.7% vs. 20.4% vs. 21.1%; *p* < 0.001) and immunotherapy (0.01% vs. 0.01% vs. 0.2% vs. 1.1%; *p* < 0.001) significantly increased from period 1 to 4. The use of laparotomy decreased significantly from 42.1% in period 2 to 16.7% in period 4, while robotic surgery usage significantly increased from 41.5% in period 2 to 64.3% in period 4. The 30-day and 90-day mortality decreased from 0.6% in period 1 to 0.2% in period 4 and 1.4% in period 1 to 0.6% in period 4, respectively. Over the period studied, we found increased use of immunotherapy, chemotherapy, and minimally invasive surgery for the management of endometrial cancer. Overall, the time interval from cancer diagnosis to final surgery increased by about 6 days. The improvements observed in the outcomes examined can probably be associated with the treatment trends observed.

## 1. Introduction

Globally, endometrial cancer is the second most common gynecological cancer after cervical cancer. It is also the most frequently diagnosed gynecological cancer in the world, with an estimated 417,000 new cases diagnosed in 2020 [1]. About 65,950 new cases of endometrial cancer occurred in the United States (US) in 2022, according to the American Cancer Society [2]. There has been an increase in the incidence of endometrial cancer in the US, mainly due to its association with aging, obesity, and diabetes [2,3]. Pre-menopausal women are currently being diagnosed with endometrial cancer at a higher rate; however, it is still primarily a disease of post-menopausal women [4,5]. The reasons why more cancers are being diagnosed in pre-menopausal women are likely multifactorial, but increasing rates of obesity and greater estrogen exposure in women who go through menarche at an earlier age may play a large role in this [6,7].

Studies [8,9,10] examining the treatment patterns and clinical outcomes for individuals with endometrial cancer have largely focused their attention on advanced or recurrent cases, neglecting the importance of early-stage endometrial cancer cases. Furthermore, some of these studies [9,10] have used study participants that are not nationally representative, leading to generalization fallacies. Using nationally representative study participants and examining both early-stage cases and advanced endometrial cancer cases together will help address some of the gaps in the literature.

Traditionally, hysterectomy, bilateral salpingo-oophorectomy, and pelvic lymph node dissection followed by adjuvant therapy based on the histological diagnosis have been the standard treatment of endometrial cancer [11]. In the last 10 to 15 years, the management of endometrial cancer has become more advanced and individualized due to several factors, including changes in histological classification that affect surgical management, adjuvant therapies, prognostic classification, and treatment approaches as well as indications for lymphadenectomy [12,13]. 

Furthermore, the integrated molecular characterization of endometrial cancer by The Cancer Genome Atlas Research Network (TCGA) has given deeper insight into the true biology of this cancer; this in turn helps to correctly match the right treatment modalities with different molecular subtypes of endometrial cancer, leading to improved outcomes [14]. The introduction of minimally invasive surgery as a treatment approach compared to open surgery has resulted in improved short- to long-term outcomes without compromising survival [15,16]. Previous studies [17,18,19] indicated that pre-operative chemotherapy and radiotherapy are beneficial in the treatment of endometrial cancer. In the last few years, total neoadjuvant chemotherapy has also been shown to be favorable in reducing involved surgical margins and in attaining a complete and full pathologic response [20]. Immunotherapy as a treatment modality in advanced endometrial cancer has also shown promising initial results [21].

As a result of recent advances in treatment approaches to endometrial cancer, we examined the National Cancer Database (NCDB) [22] over a 16-year period to assess temporal trends in the characteristics, treatment, and outcomes of endometrial carcinoma. The NCDB is a combined project of the Commission on Cancer of the American College of Surgeons and the American Cancer Society and includes hospital registry data (over 70% of all cancer diagnoses) [23] from over 1500 Commission on Cancer accredited hospitals in the US. The main aim of the present study was to determine if changes in endometrial cancer characteristics and treatment modalities were associated with a comparable significant change in outcomes over the study period.

## 2. Methods

### 2.1. Study Design and Setting

We performed retrospective analysis of the 2005 to 2020 data from NCDB of patients diagnosed with endometrial carcinoma. The Institutional Review Board (IRB) at The University of Texas Medical Branch did not consider this study human subjects research; therefore, it did not require approval.

### 2.2. Study Population

The NCDB Participant User File (PUF) was examined by two of the authors (VA and FG). Only individuals diagnosed with endometrial cancer (International Classification of Diseases for Oncology, Third Edition [ICD-O-3] codes 8000–8379, 8380, 8381–8790, 8981, and 9700–9701) and recorded in the NCDB between 2005 and 2020 were included. In addition, metastatic and non-metastatic cancer cases were eligible for inclusion, irrespective of the treatment approach adopted. Individuals with other tumor histological types including ependymoma, endometriosis, neuroendocrine tumors, sarcoma, and endometrial stromal tumors were excluded from the analysis.

### 2.3. Data Collection

Patients’ ages were categorized as <50 and ≥50. Race/ethnicity was classified as non-Hispanic white, non-Hispanic black, Hispanic, and other non-Hispanic. Insurance status was categorized as Medicaid, Medicare, private, other insurance, and uninsured. Residency was categorized as metropolitan, rural, and urban. For comorbid conditions, we examined the effect of the Charlson comorbidity score, which is based on ICD-10 codes for chronic diseases, categorized as 0, 1, 2, and ≥3. Tumor stage was categorized according to the American Joint Committee on Cancer (AJCC) 8th edition guidelines [24]. Tumor histology was defined according to the International Classification of Disease for Oncology (ICD-O) and categorized as Type I (adenocarcinoma, adenocarcinoma tubular, papillary adenocarcinoma, endometrioid, mucinous adenocarcinoma, adenocarcinoma with squamous metaplasia/adenosquamous), Type II (serous/papillary serous, clear cell), and other endometrial cancers [25]. Other factors examined included facility location (New England, Middle Atlantic, South Atlantic, East North Central, East South Central, West North Central, West South Central, Mountain Pacific), facility type (community cancer program, comprehensive community cancer program, academic/research program, integrated network cancer program), TNM (tumor, node, metastasis) stage (TNM stage was assigned the value of the reported pathologic stage and if the pathologic stage was not reported, the TNM stage was assigned the value of the clinical stage), tumor grade, lympho-vascular invasion, number of lymph nodes examined, number of positive lymph nodes harvested, administered chemotherapy and immunotherapy, sequencing of systemic therapy and radiotherapy (neoadjuvant, intraoperative, adjuvant, neoadjuvant, and adjuvant), type and approach of surgery, days from diagnosis to surgery, and first treatment as well as final treatment. The outcomes of interest included conversion of minimally invasive surgery to open surgery, 30-day and 90-day mortality, 30-day readmission, and overall survival. We subdivided the cohort into 4 consecutive periods to evaluate changes in patient demographics, tumor characteristics, treatments, and outcomes following the methodological approach of a study on rectal adenocarcinoma conducted by Emile et al. [26].

### 2.4. Data Analysis

Mean and standard deviation were used to describe all continuous variables when the data were uniformly distributed. For continuous variables that were not uniformly distributed, median and interquartile range statistics were used. Analysis of variance (ANOVA) was used to compare the means of continuous uniformly distributed variables. Chi-squared tests were used to analyze the categorical variables, which are presented as numbers and percentages. Assessment of differences in overall survival between the periods studied was conducted with Kaplan–Meier statistics and log-rank tests. SAS statistical software for Windows version 9.4 was used for all statistical analyses. A *p*-value of <0.05 was used to define statistical significance.

## 3. Results

### 3.1. Description of the Entire Cohort

We included a total of 569,817 patients diagnosed with endometrial carcinoma in our analysis after the removal of patients with other endometrial cancer histology. The mean (SD) age of the patients at diagnosis was 62.7 (11.6) years. Of the total cohort, 66,184 patients (11.6%) were younger than 50 years at diagnosis. Of the patients studied, 36,079 (6.3%) identified as Hispanics, 52,801 (9.3%) as Non-Hispanic Black, and 432,058 (75.8%) as Non-Hispanic White. Almost all patients (94.9%) resided in metropolitan or urban areas, 41.3% were insured by Medicare, and 47.2% had private insurance (Table 1). Regarding tumor type, 82.4% were Type I and 9.4% Type II, while 8.1% were of other histology types. Furthermore, 0.6% of the patients studied had TNM stage 0, 68.2% had stage I, 5.1% had stage II, 11.1% had stage III, 6.9% had stage IV disease, and 9.0% had unknown stage.

### 3.2. Patient and Tumor Characteristics across Periods Studied

Patients were subdivided into four periods: period 1 (2005–2008), period 2 (2009–2012), period 3 (2013–2016), and period 4 (2017–2020). Period 1 had 106,955 patients; period 2 had 131,786 patients; period 3 had 158,942 patients; and period 4 had 172,134 patients. The mean (SD) age at diagnosis of endometrial carcinoma increased from 62.3 (12.2) years in the first period to 63.3 (11.4) years in the last period. (Table 1). The total number of patients with early-onset cancer (before age 50 years) increased from period 1 to 4, but in terms of percentage, there was a decrease from 13.4% (period 1) to 10.7% (period 4). From period 1 to 4, there was a slight increase in the proportion of Hispanic patients (4.8% to 7.6%), and Non-Hispanic Black patients (7.3% to 10.7%). More patients with severe comorbidities were diagnosed in period 4 (3.9% vs. ≤1.9% in the previous three periods). The percentage of patients with Medicare insurance increased, from 38.9% in period 1 to 44.7% in period 4. Similarly, the percentage of patients with Medicaid coverage also increased from 4.2% in period 1 to 7.2% in period 4 (Table 1). Fewer patients in period 4 presented with locally advanced (stage III) or metastatic (stage IV) disease compared to the previous periods. Patients in the 3rd period (2013–2016) had the highest percentage of stage IV cancer diagnoses (4.5% vs. 2.9% vs. 4.1% vs. 1.2%; *p* < 0.001).

A reduction in the proportion of patients diagnosed with Type I (85.9% to 80.6%), and an increase in the proportion of patients diagnosed with Type II tumors (7.4% to 11.6%) was noted from period 1 to period 4. The median (IQR) number of lymph nodes examined decreased from six (0–16) in the first period to three (0–8) in the last period (Table 1).

### 3.3. Temporal Trends in Treatment Approaches

Use of systemic chemotherapy (14.1% vs. 17.7% vs. 20.4% vs. 21.1%; *p* < 0.001) (Table 2 and Figure 1) and immunotherapy (0.01% vs. 0.01% vs. 0.2% vs. 1.1%; *p* < 0.001) (Table 2 and Figure 2) significantly increased across the four periods studied. Neoadjuvant systemic therapy and adjuvant systemic therapy use increased from period 1 to period 4 (Table 2). The use of combined neoadjuvant and adjuvant systemic therapy was more frequent over time (Table 2 and Appendix A). The approach to surgery significantly changed as the use of laparotomy decreased significantly from 42.1% in period 2 to 16.7% in period 4. Conversely, the use of robotic surgery increased from 41.5% in period 2 to 64.3% in period 4) (Table 2 and Figure 3).

### 3.4. Temporal Trends in Outcomes

The surgical conversion rate from minimally invasive surgery to open surgery/laparotomy was slightly reduced (1.9% in period 2 to 1.8% in period 4) (Table 2). Similarly, the length of hospital stays for admitted patients was reduced by 2 days, from 3 days in period 1 to 1 day in period 4 (Table 2). There was a significant reduction in the rates of 30-day mortality (0.6% to 0.2%; *p* < 0.001) and 90-day mortality (1.4% to 0.6%; *p* < 0.001) and 30-day planned readmission (1.6% to 0.7%; *p* < 0.001) for patients who underwent surgery (Table 2). The 5-year survival rate increased from 78.4% in the first period to 79.7% and 80.8% in the following two periods, respectively. The last period was not included as it did not have a long enough length of follow-up. Overall, there was a statistically significant difference in survival among the first three periods (log-rank < 0.001, Kaplan–Meier graph) (Figure 4). The reported log-rank test is a global test of differences in survival among any of the first three periods.

## 4. Discussion

We examined trends in patients’ characteristics, treatments, and outcomes of endometrial carcinoma across the US from 2005 through 2020. The current study demonstrated a 60.9% increase in the overall incidence of endometrial carcinoma over the study period. Furthermore, there has been a 28.6% increase in the incidence of early-onset (younger than 50 years old) endometrial cancer over the study period. Previous studies have shown that the incidence of endometrial carcinoma has been rising faster than that of other gynecological cancers in the US, with an annual incidence of 2% in women younger than 50 years and by 1% in women older than 50 years from the 1990s to 2020 [24,27,28]. Available evidence from the Surveillance, Epidemiology, and End Results (SEER) database indicates a sustained rise in the incidence of early-onset uterine cancer in the US between 1991 and 2019 [29]. Moreover, the percentage of individuals diagnosed with endometrial carcinoma in period 4 that had multimorbidity was more than three times that in period 1. This observation could attest to the availability of advanced healthcare delivery services, permitting comprehensive and better management of individuals living with severe comorbidities [30].

We observed significant changes in the treatment approach to endometrial carcinoma over the 16-year interval we examined. Chemotherapy use significantly increased by 1.5 times over the study periods. A randomized controlled trial by Randall et al. [31] revealed that doxorubicin–cisplatin (AP) chemotherapy administered to women with stage III or IV endometrial carcinoma significantly improved progression-free and overall survival.

In the same vein, immunotherapy use significantly increased by about 110 times over the study period. The use of immunotherapy in treating endometrial carcinoma has been described as a game changer and promising alternative for the treatment of advanced and recurrent endometrial carcinoma [32]. In immunotherapies, an individual’s own immune system is used to fight cancer, to pave the way for more specific and effective treatments. Immunotherapy use results in fewer side effects when compared to chemotherapy [33]. Two randomized clinical trials recently published in the New England Journal of Medicine (NEJM) found immunotherapy added to standard treatment for advanced and recurrent endometrial cancer in people with stage III and IV disease improves progression-free survival [34,35].

Over the years, the surgical management of endometrial carcinoma has undergone significant transformation in the US as the use of open surgery decreased remarkably in favor of assisted minimally invasive surgery. This finding could be the result of enhanced performance of endometrial carcinoma surgery in most specialized centers due to the non-requirement of a large abdominal incision, less postoperative pain, shorter hospital stays, faster postoperative recovery, improved cosmetic outcomes, and lower costs [36,37]. The results of the present study support those of similar studies [16,38] that have compared perioperative outcomes of the three surgical approaches for endometrial carcinoma.

Another noticeable finding in this study was the increase in the use of robotic surgery by 22.8% (from 41.5% in period 2 to 64.3% in period 4), which is conversely consistent with the reduction in the use of open surgery. Minimally invasive surgery methods including robotic surgery and laparoscopic surgery are often preferred as they result in less blood loss, shorter hospital stays, and fewer postoperative complications compared to open surgery (laparotomies) [39,40,41].

An increase in time duration from diagnosis to first and final treatment was statistically significant across all of the time periods studied. Overall, the median time between diagnosis and first treatment was below 42 days, which is consistent with the findings of previous studies [42,43]. Furthermore, the median wait time of 6 weeks in the present study is in tandem with the benchmark wait time for endometrial surgical treatment in Canada as recommended by Cancer Care Ontario (CCO) [44]. To the best of our knowledge, no benchmark wait time currently exists in the US for timely surgery of endometrial cancer. These results, which may be applicable to other cancers, highlight the significance of having national standards for surgical wait times to optimize overall positive outcomes in individuals diagnosed with cancer. The significant improvement in survival and other outcomes of interest seen in this research work, despite the increased cases of endometrial cancer over the study period, could be because of better multidisciplinary care and improved therapeutic options. The establishment of a National Accreditation Program for endometrial carcinoma, like breast cancer [45], may shed more light on the outcome trends observed in the future.

### Strengths and Limitations

Very little has been documented in the literature regarding changes in treatment patterns and outcomes of endometrial cancer, which is a major strength of this research work. Findings from this study should be interpreted in the context of its limitations. One limitation of this study as with most retrospective studies is the use of existing data, which provided us with limited data to examine additional factors including but not limited to obesity, diabetes, and the recurrence of cancer after treatment. Second, using a very large sample size of NCDB data may bring about a type I error because some statistically significant findings may not have clinical relevance. Third, case series studies do not have comparison groups, making causal inferences difficult to establish. Fourth, NCDB includes only data that are hospital-based compared to other population-based databases that include both hospital-based and community-based data found in some countries, which is not currently available in the US, and therefore may not be fully representative of the overall US endometrial cancer population. Thus, generalizability from the data used for this study is limited when compared to other databases having both hospital- and community-based data available in other settings. Fifth, we reported overall survival as we were not able to obtain data to verify if the cause of death in those who died was cancer-specific or not, which may not be a true reflection of survival after treatment of endometrial carcinoma. Sixth, the findings of this study should be taken into consideration within the given population context as most of the patients analyzed were non-Hispanic white and almost all resided in metropolitan areas. Seventh, it is unclear whether the improved survival seen in this study was due to fewer advanced stages (due to early detection) when compared to early stages or advances in treatment approaches. To evaluate this accurately, we need to look at the difference in survival by the stage of the cancer. Eighth, we were not able to examine the impact of molecular subtyping on treatment planning and associated outcomes as molecular subtyping data is not available for use in the NCDB database. Lastly, future large-scale studies are needed that will examine the associations between obesity, diabetes, parity, hormone use, and endometrial cancer to provide insight into why endometrial cancer rates are increasing in the US.

## 5. Conclusions

We found increasing trends in the use of chemotherapy, immunotherapy, and minimally invasive surgery for the management of individuals with endometrial cancer in the US over a period of 16 years. Moreover, the time interval between the cancer diagnosis and final surgery increased significantly. The changes in treatment patterns seen in this study are likely linked to the significant improvements in survival, hospitalization, and readmission. The results of this study are of vital clinical significance to the holistic management of endometrial carcinoma.

## Figures and Tables

**Figure 1 cancers-16-01282-f001:**
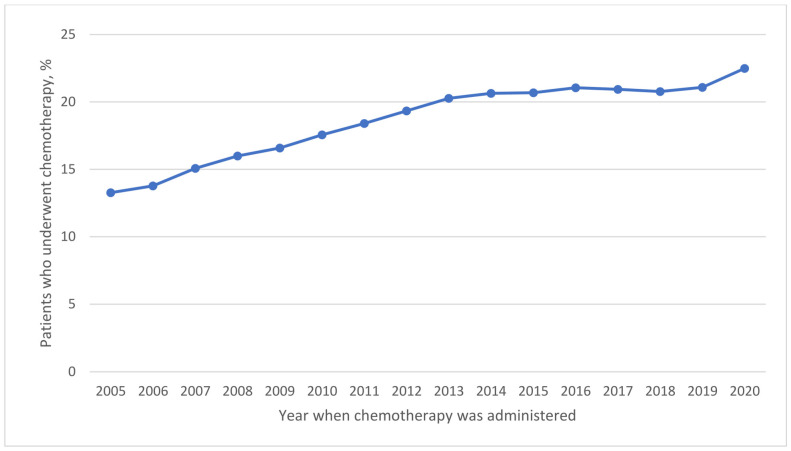
Trends in the Use of Chemotherapy (2005–2020).

**Figure 2 cancers-16-01282-f002:**
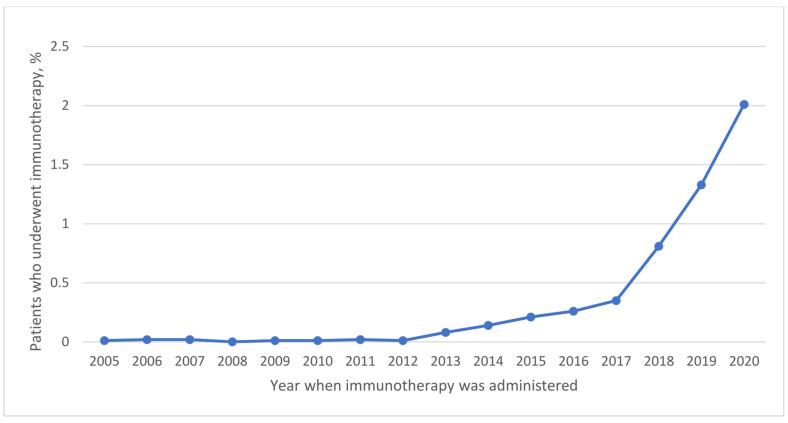
Trends in the Use of Immunotherapy (2005–2020).

**Figure 3 cancers-16-01282-f003:**
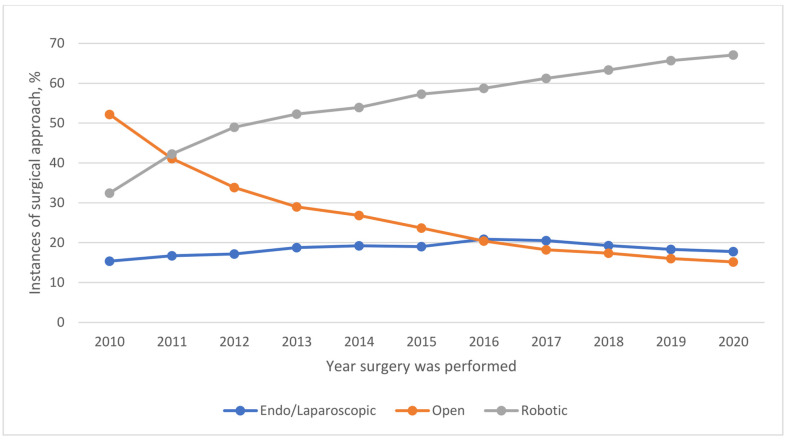
Trends in the Surgical Approach Used (2010–2020).

**Figure 4 cancers-16-01282-f004:**
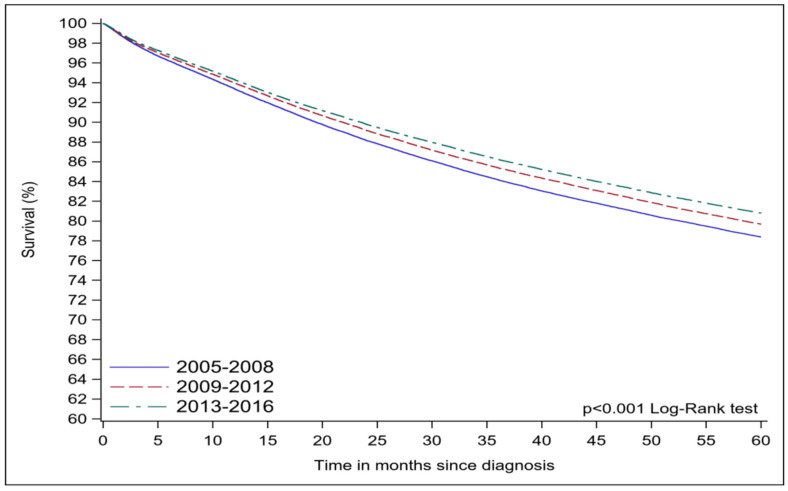
5-year overall survival.

**Table 1 cancers-16-01282-t001:** Patient Characteristics and Pathologic Parameters Across 4 Periods (2005–2020).

Characteristics	No. (%)				
	2005–2008 (*n* = 106,955)	2009–2012 (*n* = 131,786)	2013–2016 (*n* = 158,942)	2017–2020 (*n* = 172,134)	*p*-Value
Age, Mean (SD)	62.3 (12.2)	62.4 (11.7)	62.6 (11.4)	63.3 (11.4)	<0.001
Age Category					<0.001
<50	14,302 (13.4)	15,782 (12.0)	17,710 (11.1)	18,390 (10.7)	
≥50	92,653 (86.6)	116,004 (88.0)	141,232 (88.9)	153,744 (89.3)	
Race and Ethnicity					<0.001
Hispanic	5142 (4.8)	7423 (5.6)	10,406 (6.6)	13,108 (7.6)	
Non-Hispanic Black	7836 (7.3)	11,345 (8.6)	15,222 (9.6)	18,398 (10.7)	
Non-Hispanic White	80,178 (75.0)	101,613 (77.1)	122,224 (76.9)	128,043 (74.4)	
Other non-Hispanic	13,799 (13.0)	11,405 (8.7)	11,090 (7.0)	12,585 (7.3)	
Charlson Comorbidity Index Score					<0.001
0	80,062 (74.9)	96,650 (73.3)	116,458 (73.3)	125,700 (73.0)	
1	21,285 (19.9)	27,654 (21.0)	32,128 (20.2)	30,998 (18.0)	
2	4290 (4.0)	5750 (4.4)	7287 (4.6)	8811 (5.1)	
3	1318 (1.2)	1732 (1.3)	3069 (1.9)	6625 (3.9)	
Residency					<0.001
Metropolitan	86,502 (80.9)	107,147 (81.3)	129,746 (81.6)	141,874 (82.4)	
Rural	1823 (1.7)	2193 (1.7)	2595 (1.6)	2643 (1.5)	
Urban	14,009 (13.1)	17,258 (13.1)	21,207 (13.3)	22,872 (13.3)	
Insurance Status					<0.001
Medicare	41,625 (38.9)	51,538 (39.1)	65,073 (40.9)	76,950 (44.7)	
Medicaid	4461 (4.2)	6718 (5.1)	10,331 (6.5)	12,394 (7.2)	
Other	2799 (2.6)	3522 (2.7)	4013 (2.5)	3492 (2.0)	
Private	54,589 (51.0)	64,780 (49.2)	74,836 (47.1)	74,892 (43.5)	
Uninsured	3481 (3.3)	5228 (4.0)	4689 (3.0)	4406 (2.6)	
Facility Location					<0.001
New England	7151 (6.9)	7970 (6.3)	9909 (6.4)	10,005 (6.0)	
Middle Atlantic	18,286 (17.7)	22,767 (17.8)	26,762 (17.4)	27,866 (16.8)	
South Atlantic	19,823 (19.2)	24,954 (19.6)	29,777 (19.4)	33,464 (20.1)	
East North Central	20,228 (19.6)	23,425 (18.4)	27,037 (17.6)	28,762 (17.3)	
East South Central	6290 (6.1)	7684 (6.0)	9320 (6.1)	10,029 (6.0)	
West North Central	8869 (8.6)	11,245 (8.8)	13,309 (8.7)	13,598 (8.2)	
West South Central	6841 (6.6)	9012 (7.1)	11,762 (7.6)	13,383 (8.1)	
Mountain	4286 (4.2)	5398 (4.2)	6357 (4.1)	6834 (4.1)	
Pacific	11,613 (11.2)	15,130 (11.9)	19,637 (12.8)	22,349 (13.4)	
Facility Type					<0.001
Community Cancer Program	4766 (4.6)	4990 (3.9)	5313 (3.5)	5807 (3.5)	
Comprehensive Community Cancer Program	37,507 (36.3)	45,759 (35.9)	54,589 (35.5)	59,463 (35.8)	
Academic/Research Program	38,484 (37.2)	49,240 (38.6)	61,759 (40.1)	65,865 (39.6)	
Integrated Network Cancer Program	22,630 (21.9)	27,596 (21.6)	32,209 (20.9)	35,155 (21.1)	
TNM stage					<0.001
0	1119 (1.0)	1190 (0.9)	1033 (0.6)	187 (0.1)	
1	69,345 (64.8)	92,059 (69.9)	110,698 (69.6)	116,606 (67.7)	
2	8025 (7.5)	6994 (5.3)	7081 (4.5)	6857 (4.0)	
3	12,471 (11.7)	14,435 (10.9)	17,594 (11.1)	18,935 (11.0)	
4	5533 (5.2)	7099 (5.4)	9415 (5.9)	12,178 (7.1)	
Unknown	10,462 (9.8)	10,009 (7.6)	13,121 (8.3)	17,371 (10.1)	
Grade					<0.001
Well differentiated	43,603 (40.8)	51,106 (38.8)	56,288 (35.4)	75,931 (44.1.)	
Moderately differentiated	31,057 (29.0)	34,175 (25.9)	32,684 (20.6)	38,801 (22.5)	
Poorly differentiated	20,215 (18.9)	23,110 (17.5)	24,249 (15.3)	28,882 (16.8)	
Undifferentiated	2375 (2.2)	3913 (3.0)	5871 (3.7)	1944 (1.1)	
Unknown	9705 (9.1)	19,482 (14.8)	39,850 (25.1)	26,576 (15.4)	
Lymphovascular Invasion					<0.001
No	NA	67,069 (50.9)	107,541 (67.7)	113,151 (65.7)	
Yes	NA	16,787 (12.8)	27,509 (17.3)	32,478 (18.9)	
Unknown	NA	47,930 (36.3)	23,892 (15.0)	26,505 (15.4)	
Type of Histology					<0.001
Type I	91,881 (85.9)	109,614 (83.2)	129,405 (81.4)	138,804 (80.6)	
Type II	7937 (7.4)	10,450 (7.9)	15,502 (9.8)	19,883 (11.6)	
Others	7137 (6.7)	11,722 (8.9)	14,035 (8.8)	13,447 (7.8)	
No. of Lymph Nodes Examined, Median (IQR)	6 (0–16)	6 (0–16)	4 (0–14)	3 (0–8)	<0.001
No. of Positive Lymph Nodes, Median (IQR)	0 (0–1)	0 (0–1)	0 (0–1)	0 (0–1)	<0.001

Abbreviations: NA, not available; TNM, tumor, node, metastasis. TNM stage is based on NCDB’s best stage variable, which represents a combination of American Joint Committee on Cancer (AJCC) pathologic stage over clinical stage. In this cohort, 60.9% of TNM cancer stage was from pathologic stage. Number and percentages may not add up because of missing data.

**Table 2 cancers-16-01282-t002:** Treatment and Outcome Trends Across Four Periods (2005–2020).

Factor	2005–2008	2009–2012	2013–2016	2017–2020	*p*-Value
Chemotherapy					<0.001
No	88,193 (82.5)	105,895 (80.4)	124,495 (78.3)	134,004 (77.9)	
Yes	15,080 (14.1)	23,293 (17.7)	32,449 (20.4)	36,255 (21.1)	
Immunotherapy					<0.001
No	104,577 (97.8)	131,003 (99.4)	158,314 (99.6)	169,846 (98.7)	
Yes	15 (0.01)	17 (0.01)	278 (0.2)	1901 (1.1)	
Sequencing of Systemic Therapy					<0.001
Adjuvant	10,865 (10.2)	21,023 (16.0)	28,243 (17.8)	30,138 (17.5)	
Intraoperative	5 (0.00)	26 (0.02)	27 (0.02)	49 (0.03)	
Neoadjuvant	518 (0.5)	966 (0.7)	1713 (1.1)	2307 (1.3)	
Neoadjuvant and Adjuvant	156 (0.2)	457 (0.4)	1067 (0.7)	1883 (1.1)	
No Treatment	68,355 (63.9)	107,182 (81.3)	126,500 (79.6)	136,444 (79.3)	
Unknown	27,056 (25.3)	2132 (1.6)	1392 (0.9)	1313 (0.8)	
Sequencing of Radiotherapy					<0.001
Adjuvant	24,605 (23.0)	28,567 (21.7)	37,668 (23.7)	44,548 (25.9)	
Intraoperative	9 (0.01)	10 (0.01)	12 (0.01)	6 (0.0)	
Neoadjuvant	542 (0.5)	534 (0.4)	719 (0.5)	839 (0.5)	
Neoadjuvant and Adjuvant	53 (0.1)	59 (0.04)	82 (0.1)	104 (0.1)	
No Treatment	77,347 (72.3)	98,349 (74.6)	116,348 (73.2)	123,856 (71.9)	
Unknown	4399 (4.1)	4267 (3.2)	4113 (2.6)	2781 (1.6)	
Type of Surgery					<0.001
Local tumor destruction/excision	1078 (1.0)	1281 (1.0)	1567 (1.0)	1574 (0.9)	
Total hysterectomy	1310 (1.2)	1663 (1.3)	1680 (1.1)	1635 (1.0)	
Radical hysterectomy	84,013 (78.6)	104,289 (79.1)	127,842 (80.4)	140,365 (81.5)	
Hysterectomy and Pelvic exenteration	12,602 (11.8)	14,820 (11.3)	15,441 (9.7)	13,460 (7.8)	
Total	106,955	131,786	158,942	172,134	
Approach of Surgery					<0.001
Endo or Laparoscopic	NA	14,049 (16.4)	26,004 (19.5)	26,794 (19.0)	
Open	NA	35,989 (42.1)	33,079 (24.8)	23,587 (16.7)	
Robotic	NA	35,437 (41.5)	74,207 (55.7)	90,632 (64.3)	
Total	NA	85,475	133,290	141,013	
Conversion					<0.001
No	NA	83,002 (63.0)	129,793 (81.7)	137,928 (80.1)	
Yes	NA	2473 (1.9)	3497 (2.2)	3085 (1.8)	
Total	NA	85,475	133,290	141,013	
30 d Mortality					<0.001
No	97,776 (91.4)	120,608 (91.5)	144,893 (91.2)	118,189 (68.7)	
Yes	657 (0.6)	683 (0.5)	583 (0.4)	373 (0.2)	
Total	98,433	121,291	145,476	118,562	
90 d Mortality					<0.001
No	96,710 (90.4)	119,404 (90.6)	143,357 (90.2)	116,361 (67.6)	
Yes	1467 (1.4)	1555 (1.2)	1488 (0.9)	990 (0.6)	
Total	98,177	120,959	144,845	117,351	
30 d Readmission					<0.001
No readmission	96,933 (90.6)	124,553 (94.5)	152,322 (95.8)	166,099 (96.5)	
Planned readmission	1693 (1.6)	1583 (1.2)	1520 (1.0)	1235 (0.7)	
Unplanned readmission	3533 (3.3)	3439 (2.6)	3354 (2.1)	2986 (1.7)	
Total	102,159	129,575	157,196	170,320	
Hospital Stay, Median (IQR)	3 (2–4)	2 (1–3)	1 (1–2)	1 (0–1)	<0.001
Time from Diagnosis to First Surgery, Median (IQR)	24 (1–40)	27 (5–42)	28 (10–44)	30 (13–47)	<0.001
Time from Diagnosis to Final Surgery, Median (IQR)	26 (10–41)	28 (12–43)	30 (15–46)	32 (16–48)	<0.001
Time from Diagnosis to First Treatment, Median (IQR)	24 (3–40)	27 (6–42)	28 (10–44)	29 (13–46)	<0.001

Abbreviations: NA, not available. Number and percentages may not add up because of missing data.

## Data Availability

Restrictions apply to the availability of the datasets used for this study. The NCDB database is controlled by the ACS and is available upon request and approval. Requests to access the database should be directed to postmaster@facs.org. The methods for our analysis will be made available at request.

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
