# Peer review of "Temporal Trends in Treatment and Outcomes of Endometrial Carcinoma in the United States, 2005–2020"

_cancers, 2024, doi:10.3390/cancers16071282_

Round 1
Reviewer 1 Report
Comments and Suggestions for Authors
The Cancer Genome Atlas Research Network subdividing endometrial cancer was published one decade ago. Since then a deeper insight into true cancer biology has reached what in turn allows to fit correct regiment and predict an outcome.
Unfortunately, this aspect is missed in review.
Secondly, the Authors inform us that all data were taken from the National Cancer Database. Table 1 presents many parameters where crucial data such as cTNM, pTNM, Grade, LVI are incomplete ( line ‘unknown’). The number of missed data makes all statistics incorrect and cannot be accepted.
Reviewer 2 Report
Comments and Suggestions for Authors
The abstract effectively summarizes the study's objectives, methodology, key findings, and conclusions. However, it would benefit from more specific numerical data regarding observed trends to enhance clarity and provide readers with a clearer understanding of the study's scope and implications.
The introduction provides comprehensive background information on the epidemiology, risk factors, and traditional treatment approaches of endometrial carcinoma. It effectively contextualizes the study within the broader landscape of gynecologic cancers and highlights the need for research on temporal trends in treatment and outcomes. The introduction appropriately justifies the study's importance and sets the stage for further investigation. Strengthen the rationale for the study by highlighting the gap in existing literature that the research aims to address.
The methods section outlines the study design, data source, inclusion criteria, variables examined, and statistical analyses conducted. It provides a clear and detailed description of the methodology, enabling replication and understanding of the study procedures. The use of the National Cancer Database (NCDB) and the rationale for subdividing patients into four periods are appropriately justified. However, discussing potential biases or limitations associated with using retrospective data from the NCDB could enhance the transparency of the study.
The results section presents the findings of the study in a structured manner, addressing patient characteristics, treatment approaches, and outcomes across the four study periods. The use of descriptive statistics, tables, and figures effectively communicates the observed trends. The results are supported by numerical data and statistical analyses, enhancing the credibility of the findings. However, you should provide additional context for certain observed trends, especially regarding their implications for clinical practice, would strengthen the discussion.
The discussion section thoroughly interprets the study findings in the context of existing literature, clinical implications, and potential limitations. It effectively highlights the significance of observed trends in treatment approaches and outcomes of endometrial carcinoma. The discussion provides critical insights into the evolving landscape of endometrial cancer management and underscores the importance of further research to address remaining gaps or limitations. Strengthening explicit connections between observed trends and their implications for patient care, healthcare policy, or future research directions would enhance the discussion's impact.
Overall, the manuscript provides valuable insights into temporal trends in endometrial carcinoma treatment and outcomes in the United States. The study is well-conducted, and the findings are presented clearly and comprehensively. Addressing minor suggestions for improvement would enhance the manuscript's clarity and impact.
Comments on the Quality of English LanguageAppropriate language
Reviewer 3 Report
Comments and Suggestions for Authors
1. "Abstract : The mean (SD) age at diagnosis 34 was 62.7 (11.6) years, but 66, 184 patients (11.6%) were younger than 50 years indicating that more 35 patients are getting diagnosed at younger ages."
Looking at Table 1, it seems that the more recent the incidence, the higher the age. However, the above statement seems to contradict this and needs further explanation.
2. Table 1.
In Table 1, the No. of positive lymph nodes is 0. I don't understand why the average number of positive nodes is 0 when there were many patients with stage 3-4 included.
What is the reason for the sudden increase of unknown patients of unknown stage or grade in Period 4? This seems to have a very serious impact on the reliability of the data.
3. Temporal Trends in Treatment Approaches
"The time from when cancer diagnosis was made to when 178 final surgery was done increased from 26 days in period 1 to 32 days in period 4"
What's the big deal about the extension from 26 to 32 days in a study based on big data? I'm not sure it should be in the results.
4. What do First surgery and Final surgery mean in Table 2?
5.Figure 4 shows that survival has improved in recent years, but it is unclear whether this is due to fewer advanced stages (due to early detection) or advances in adjuvant therapies. To see this accurately, we need to look at the difference in survival by stage.
As mentioned earlier, there are too many unknown stages, so the reliability of survival data seems to have serious errors. I would like the authors' opinion on this..,
Comments on the Quality of English LanguageThe overall quality of the English language in this document is commendable. The text demonstrates a high level of clarity, coherence, and grammatical accuracy, which facilitates easy understanding and engagement. The vocabulary used is appropriate and contributes to the effective conveyance of the study's findings and implications. Such proficiency in language not only enhances readability but also ensures that the scientific content is accessible to a broad audience. This level of English quality significantly contributes to the document's professionalism and the effective communication of its key points.
Round 2
Reviewer 1 Report
Comments and Suggestions for Authors Dear Authors, This review covers the 2005-2020 period. I still think that molecular subtyping is crucial for correct treatment planning and predict of outcome. Because the mentioned molecular subtyping became a gold standard over 5-6 years ago you should clearly explain why you did not implement these data. This statement should be put in the Material section. Moreover, in Table 1 the last line concerns tumor size - this parameter does not matter in endometrial cancer TNM, the local range of cancer invasion is crucial. Please remove that. I mean, you should mention that NCDB is not perfect and could lead to confusion.Author Response
This review covers the 2005-2020 period. I still think that molecular subtyping is crucial for correct treatment planning and predict of outcome. Because the mentioned molecular subtyping became a gold standard over 5-6 years ago you should clearly explain why you did not implement these data. This statement should be put in the material section.
Reply: We thank this reviewer for their thorough review of our manuscript. Based on your comment, we have added that we were not able to examine the impact of molecular subtyping on treatment planning and associated outcomes to the limitation section of this study. NCDB does not collect molecular subtyping data in their database. (see page 12, lines 312-315).
Moreover, in Table 1 the last line concerns tumor size - this parameter does not matter in endometrial cancer TNM, the local range of cancer invasion is crucial. Please remove that. I mean, you should mention that NCDB is not perfect and could lead to confusion.
Reply: Thank you for your suggestion. We have removed the last line in Table 1 as requested by this reviewer.
Round 3
Reviewer 1 Report
Comments and Suggestions for Authors
Dear Authors,
In the present form your manuscript looks better. I appreciate an effort you put to prepare this review.
My decision is 'accept'.